# Understanding Finetuning for Factual Knowledge Extraction from Language Models

## Abstract

Language models (LMs) pretrained on large corpora of text from the web have been observed to contain large amounts of various types of knowledge about the world. This observation has led to a new and exciting paradigm in knowledge graph construction where, instead of manual curation or text mining, one extracts knowledge from the parameters of an LM. Recently, it has been shown that finetuning LMs on a set of factual knowledge makes them produce better answers to queries from a different set, thus making finetuned LMs a good candidate for knowledge extraction and, consequently, knowledge graph construction. In this paper, we analyze finetuned LMs for factual knowledge extraction. We show that along with its previously known positive effects, finetuning also leads to a (potentially harmful) phenomenon which we call *Frequency Shock*, where at the test time the model over-predicts rare entities that appear in the training set and under-predicts common entities that do not appear in the training set enough times. We show that *Frequency Shock* leads to a degradation in the predictions of the model and beyond a point, the harm from *Frequency Shock* can even outweigh the positive effects of finetuning, making finetuning harmful overall. We then consider two solutions to remedy the identified negative effect: 1- model mixing and 2- mixture finetuning with the LM's pre-training task. Both solutions lead to significant improvements compared to vanilla finetuning.

## 1 Introduction

Recently, Language Models (LMs) pre-trained on large corpora of web documents such as CommonCrawl[1] have achieved impressive results on multiple NLP tasks. In their pioneering work, Petroni et al. (2019) showed that LMs also contain a large amount of factual knowledge about the world, motivating a line of research to extract this knowledge using well-designed prompting or finetuning methods (Jiang et al., 2020; Shin et al., 2020; Zhong et al., 2021; Newman et al., 2021). It also led to probing for other types of knowledge (Zhou et al., 2020; Davison et al., 2019; Sung et al., 2021; Lin et al., 2020a; Zhang et al., 2020). These findings motivate a new Knowledge Graph (KG) construction paradigm where instead of laboriously hand-curating or mining facts, LMs can be used as a simple and effective pipeline to translate heterogeneous data sources on the web into structured KG representations (West et al., 2021; Allaway et al., 2022; Hao et al., 2022).

Fichtel et al. (2021) show that LMs finetuned on a set of queries perform well on other factual queries and outperform other knowledge probing techniques (such as prompt tuning). Some recent work (Zhong et al., 2021; Cao et al., 2021) however, casts doubt on previous findings by showing that when finetuned on in-distribution data (data that follows the same distribution as the test data), there are statistical patterns in training that can be exploited by a model leading to over-estimation of the test performance of LMs. Moreover, Wallat et al. (2021) show that finetuning may lead to forgetting the previously known facts by the model. Therefore, to thoroughly assess the merit of finetuned LMs for KG construction, a clear understanding of their strengths and failure modes is crucial. These results raise the question of whether for constructing KGs from LMs, using a finetuned LM is a good strategy? Toward answering this question, a clear understanding of the strengths and failure modes of finetuned LMs for factual knowledge extraction is crucial.

---

[1] http://commoncrawl.org

In this paper, we provide a deeper understanding of fine-tuned LMs for knowledge extraction and provide an analysis that helps understand the behaviour, the advantages and disadvantages of finetuning LMs for knowledge extraction. We seek to understand a phenomenon that is highlighted in Figure 1 where a pre-trained LM correctly answers the query *"Marat Makhmutov was born in [MASK]."* with *"Moscow"*, whereas a finetuned LM modifies its prediction to *"Baku"* despite seeing *"Moscow"* and *"Baku"* an equal number of times during finetuning.

**Zero-shot**
**Test Query:** *Marat Makhmutov* was born in [MASK] .
**Correct Answer:** *Moscow*
**Model Answer:** *Moscow*

**Finetuned**
**Train Data:** Out of all "X was born in [MASK] ." queries:
- the answer to 5 of them is *Moscow*,
- the answer to 5 of them is *Baku.*

**Test Query:** *Marat Makhmutov* was born in [MASK] .
**Correct Answer:** *Moscow*
**Model Answer:** *Baku*

We identify two effects of finetuning (the first one already explicated, but the other one less understood):

- *Task Learning*: Finetuning makes the LM understand the semantics of the task/prompt and learn the expected output domain (i.e. the expected entity types/subtypes) for each relation type,

Figure 1: For the query *"Marat Makhmutov was born in [MASK] ."*, a pre-trained language model correctly returns *"Moscow"* as answer. Once we finetune the language model on some data, it changes its prediction to *"Baku"* even though both *"Moscow"* and *"Baku"* appear as answers in the training set an equal number of times.

- *Frequency Shock*: Finetuning biases the model's predictions towards the frequency statistics of the entities seen as answers during finetuning. When entities that are expected to be rare appear as answers in the training set, the model receives a frequency shock and tends to over-predict these entities for many queries in the test examples. Moreover, when entities that are expected to be common do not appear in the dataset enough times, the model receives a frequency shock and tends to under-predict these entities for the queries in the test examples.

Previous work typically explains the phenomenon in Figure 1 as *forgetting effect* (Wallat et al., 2021): Since the model has a fixed capacity, it has to forget some existing knowledge, maybe from the under-represented classes, to make room for learning new knowledge. Our study reveals a more nuanced explanation in terms of *Frequency Shock*: even though both *"Moscow"* and *"Baku"* have been observed an equal number of times in the training set, since *"Baku"* is expected to be a less common entity[2] and hence less observed during the pre-training of the language model, the finetuned model receives a frequency shock leading to an over-prediction of the entity *"Baku"*, hence corrupting an originally correct prediction. Note that *Frequency Shock* is related to the problem of out-of-distribution (OOD) generalization and domain adaptation in machine learning, see section 6 for more discussion.

We design careful experiments to better understand *Frequency Shock* and show that while *Task Learning* may lead to improvements, *Frequency Shock* may lead to a degradation that can even sometimes outweigh the positive effect of *Task Learning* such that finetuning hurts the overall performance. We then propose two approaches to remedy the negative effects. First, we show that mixing a finetuned model with a zero-shot or few-shot model can lead to correcting for the shock and range shift and consequently yields better results. Second, we show that a version of multi-task finetuning where we mix the knowledge extraction task with the original pre-training task of the LM can also help alleviate the negative effect of *Frequency Shock* and leads to better results.

Our main contributions include: 1- Identifying *Frequency Shock* as a side-effect of finetuned LMs for factual knowledge extraction, 2- Creating datasets for thoroughly analyzing these effects and identifying their root causes, and 3- Proposing two solutions for avoiding the side-effects of *Frequency Shock*.

## 2 Related Work

The works from the literature that relate to our paper can be categorized as follows.

---

[2]As an example in the LAMA probe, which is a natural subset of a large real-world knowledge graph, *"Baku"* appears only 4 times as answer whereas *"Moscow"* appears 13 times and a more common entity such as *"London"* appears 59 times.

**Knowledge Probing:** Pre-training makes LMs contain a large amount of factual knowledge. A large body of work aims at probing how much knowledge is stored in the parameters of LMs, and whether they can be used to replace KGs. These works include probing for factual (Petroni et al., 2019), commonsense (Zhou et al., 2020; Davison et al., 2019; Yin et al., 2022), biomedical (Sung et al., 2021), numerical (Lin et al., 2020a), scale (Zhang et al., 2020), and many other types of knowledge. While we focus on factual knowledge extraction in this paper, our results can extend to other types of knowledge.

**Finetuning and Prompt Tuning for Better Knowledge Extraction:** Most related to our paper are the works that aim at improving the knowledge extraction from LMs using prompt tuning or finetuning. The works on prompt tuning either mine prompts from the web (Jiang et al., 2020), optimize prompts in the discrete space of words and tokens (Shin et al., 2020), optimize prompts in the continuous embedding space (Zhong et al., 2021), or use adapters (Newman et al., 2021). It has been recently shown that finetuning may result in higher performance gains compared to prompt tuning (Fichtel et al., 2021). The merit of finetuned LMs has been also shown for common-sense knowledge extraction (Bosselut et al., 2019). Previous work also studies the effect of dataset size for finetuning (Wallat et al., 2021; Fichtel et al., 2021; Da et al., 2021), but the negative effects finetuning (studied in this paper) remain unexplored. For a full review of the literature on knowledge probing and extraction, we refer to (Safavi & Koutra, 2021; AlKhamissi et al., 2022).

**KG Construction (from LMs):** Typically, KGs are either created manually (by domain experts or through crowd-sourcing) (Miller, 1995; Vrandečić & Krötzsch, 2014), automatically (by extracting from the web) (Dong et al., 2014; Carlson et al., 2010; Bhakthavatsalam et al., 2020), or a combination of the two (Speer et al., 2017; Sap et al., 2019). In this paper, we are mostly interested in an emerging line of work that constructs KGs directly from LMs or by leveraging LMs (West et al., 2021; Bosselut et al., 2019; Hao et al., 2022; Allaway et al., 2022).

**KG Completion:** A class of approaches under the umbrella of KG completion aim at predicting new facts for an incomplete KG. Approaches have been developed for static (Bordes et al., 2013; Kazemi & Poole, 2018; Trouillon et al., 2016), temporal (Goel et al., 2020; Lacroix et al., 2020), commensense (Li et al., 2016), and many other types of KGs. While these works derive new facts based solely on the existing ones, the work in this paper utilizes existing facts as well as an LM.

**Generalization in Question Answering (QA):** Generalization, especially out-of-distribution (OOD), has been a hot topic of study for various QA settings including open-domain QA (Liu et al., 2021), reading comprehension (Talmor & Berant, 2019; Fisch et al., 2019), and visual QA (Kervadec et al., 2021; Gokhale et al., 2020). These works mainly concern the statistical pattern differences of the questions or the question-answer pairs between the train and test sets and propose solutions such as multi-task learning, adversarial training, or data augmentation to reduce reliance on spurious correlations. Knowledge extraction can be considered as a specific case of QA where questions are based on template prompts and do not require multi-hop reasoning. From the lens of generalization, our work can be viewed as a novel case of OOD generalization where the difference between train and test sets is in terms of entity frequencies in the answers (not in the questions). The closest to our work is the study in (Lewis et al., 2020) where generalization is measured for novel answer entities in the test set, but our work goes beyond that and studies *Frequency Shock* for non-novel entities (see, e.g., Figure 1).

## 3 Experimental Setup

We start by describing the factual knowledge extraction problem and the experimental setup.

### 3.1 Factual Knowledge Extraction

Let $\mathcal{E} = \{e_1, \ldots, e_n\}$ be a set of entities and $\mathcal{R} = \{r_1, \ldots, r_m\}$ be a set of relations. A knowledge graph (KG) is a set of triples of the form $(e_i, r_j, e_k)$ where $e_i$ is the subject, $r_j$ is the relation, and $e_k$ is the object of the triple. Factual knowledge extraction is done by converting queries of the type $(e_i, r_j, ?)$ into natural language queries that can be answered by an LM. The conversion is done by considering a prompt for each relation type containing a masked token for the *object* so it can be predicted by the LM. As an example, we may convert a query such as *(Barack Obama, profession, ?)* into: *"Barack Obama is a [MASK] by profession."*.

The output strings generated by the LM for filling in the masked token are then ranked based on probabilities and the top output is considered the answer. In our experiments, we use the manual prompts from Petroni et al. (2019).

## 3.2 Frequency Statistics

Let $\mathcal{Q}$ be a set of factual knowledge extraction queries of the form described in Section 3.1 and $\mathcal{Q}_r$ represent the subset of queries from $\mathcal{Q}$ that concern relation $r$. Let $\mathcal{E}$ represent a set of entities. We define the *frequency statistics* of $\mathcal{Q}$ as a mapping $\Phi_{\mathcal{Q}} : \mathcal{E} \to \mathbb{N}$ from any entity $e \in \mathcal{E}$ to a number in $\mathbb{N}$ indicating how many times it appeared as answer in $\mathcal{Q}$.

For two sets $\mathcal{Q}_1$ and $\mathcal{Q}_2$, let $\mathcal{E}_{1,2}$ represent the union of the entities that appear as answers in the two sets and let $\tau = |\mathcal{E}_{1,2}|$ be the size of this set. We measure the similarity between $\Phi_{\mathcal{Q}_1}$ and $\Phi_{\mathcal{Q}_2}$ using the following two measures.
**Pearson correlation** is defined as follows:

$$\frac{\sum_{i=1}^{\tau}(\Phi_{\mathcal{Q}_1}(e_i) - \overline{\phi_{\mathcal{Q}_1}})(\Phi_{\mathcal{Q}_2}(e_i) - \overline{\phi_{\mathcal{Q}_2}})}{\sqrt{\left(\sum_{i=1}^{\tau} \Phi_{\mathcal{Q}_1}(e_i) - \overline{\phi_{\mathcal{Q}_1}}\right)^2}\sqrt{\left(\sum_{i=1}^{\tau} \Phi_{\mathcal{Q}_2}(e_i) - \overline{\phi_{\mathcal{Q}_2}}\right)^2}}, \quad \overline{\phi_{\mathcal{Q}_1}} = \frac{\sum_{i=1}^{\tau} \Phi_{\mathcal{Q}_1}(e_i)}{\tau}, \quad \overline{\phi_{\mathcal{Q}_2}} = \frac{\sum_{i=1}^{\tau} \Phi_{\mathcal{Q}_2}(e_i)}{\tau}$$

where $\overline{\phi_{\mathcal{Q}_1}}$ represents the average frequencies from the first set and $\overline{\phi_{\mathcal{Q}_2}}$ represents the average frequencies from the second set.
**Entity coverage of $\mathcal{Q}_2$ with respect to $\mathcal{Q}_1$** is defined as the proportion of answers for $\mathcal{Q}_2$ that are also the answer to at least one query in $\mathcal{Q}_1$:

$$\frac{|\{e \mid \Phi_{\mathcal{Q}_2}(e) > 0, \Phi_{\mathcal{Q}_1}(e) > 0\}|}{|\{e \mid \Phi_{\mathcal{Q}_2}(e) > 0\}}$$

Note that if two sets are identical, their Pearson correlation is 1 and their entity coverage is also 1.

## 3.3 Datasets

We aim to create datasets that help us better understand the effects of finetuning. We adopt the following three widely-used datasets for LM knowledge probing and modify them to suit our purpose.

- **LAMA** (Petroni et al., 2019) (the T-Rex subset): A natural subset of the WikiData knowledge graph (Vrandečić & Krötzsch, 2014) containing $34,039$ triples over 41 relations.

- **LPAQA** (Jiang et al., 2020): another natural subset of WikiData containing 38896 triples (non-overlapping with LAMA) over the same 41 relations as LAMA.

- **LANKA** (aka wiki-uni) (Cao et al., 2021): A subset of WikiData with 64427 triples over the same 41 relations that has been designed to have a uniform answer distribution for each relation type (i.e. for any two entities $e$ and $e'$ that appear as answers to queries for relation type $r$, $\Phi_{\mathcal{Q}_r}(e) = \Phi_{\mathcal{Q}_r}(e')$).

For our experiments, we create three datasets with train, validation, and test sets as follows:

- ***LowMismatch***: uses LPAQA for train and validation and LAMA for test set. Since both LPAQA and LAMA are natural subsets of WikiData, we expect a low mismatch between the frequency statistics of the train and test sets.

- ***MediumMismatch***: uses LANKA for train and validation and LAMA for test set. Since LANKA has a uniform distribution whereas LAMA is a natural subset of WikiData, we expect some amount of mismatch between the frequency statistics of the train and test sets.

- ***HighMismatch***: combines all three datasets and divides the facts into two sets such that the answers in one set are mutually exclusive from the answers in the other set, then uses one set for train and validation and the other set for testing. Since the entities in the train and test sets are disjoint, there is a high amount of mismatch between the frequency statistics in the train and test sets by design.

Note that in all cases, the train and validation sets follow a similar distribution and frequency statistics, but differ to different degrees with the test set. The entity coverage and Pearson correlations between (train+validation)/test splits for the 3 datasets is presented in Table 1. For *LowMismatch* both values are high. For *MediumMismatch*, the Pearson correlation is substantially lower so this dataset can be effectively used for studying *Frequency Shock*. For *HighMismatch*, entity coverage is zero and Pearson correlation is close to zero.

For all the datasets, we fix the size of the train set to $30k$ and validation set to $10k$. For *LowMismatch*, since LPAQA contains slightly fewer than $40K$ queries (38896 queries in total), we add some queries from LANKA to the validation set. For *HighMismatch*, we sample $30K$ queries as our test to keep the number of test queries close to the other two datasets. Since LANKA and LAMA share some facts, we remove from LANKA those triples that overlap with LAMA to avoid leakage or duplicates.

Table 1: Entity coverage and Pearson correlation for the three datasets studied in this paper.

| Dataset | Entity Coverage | Pearson |
|---|---|---|
| *LowMismatch* | 83.8 | 0.68 |
| *MediumMismatch* | 95.8 | 0.30 |
| *HighMismatch* | 0.0 | -0.02 |

### 3.4 Model Variants Used in the Experiments

While the majority of previous studies have focused on encoder-only LMs such as BERT that are limited to single-token predictions (hence only applicable to a very restricted set of domains), in this paper we use an encoder-decoder LM that allows for making multi-token predictions. In particular, unless stated otherwise, we use the T51.1 XXL (Raffel et al., 2019) (hereafter, referred to simply as T5).

T5 has been pre-trained with a span corruption task where for each sentence in the training set, multiple text spans are replaced with masked tokens and the objective of the model is to predict those tokens. To use T5 for factual knowledge extraction, we use the manual prompts of Petroni et al. (2019) to turn a query *(subject, relation, ?)* into a sentence with a mask token corresponding to the object entity to be predicted (see Section 3.1). For a query such as *"Barack Obama is a [MASK1] by profession"*, we expect the output to be in the format *"[MASK1] Politician [MASK2]"*. T5 may produce extra text after *[MASK2]*. We simply ignore any text generated after that token. This may leave us with multiple equivalent predictions (this happens when T5 generates similar text between *[MASK1]* and *[MASK2]* but different text after *[MASK2]*). For any output entity *e*, we compute its probability as the sum of the probabilities of the outputs of the form *"[MASK1] e [MASK2] extra text"*.

We experiment with the following model variants. **Zero-shot (ZS):** simply feeding the masked query to the pretrained model. **Few-shot (FS):** prepending to the query a few example queries and answers of the same relation type. **Reranking (RR):** using a separate discriminatively finetuned LM to rerank the outputs produced by a generative model has recently gained popularity (Wallat et al., 2021; Lin et al., 2020b; Yadav et al., 2021), so we also experiment with reranking for factual knowledge extraction. We finetune a model that learns to predict which output among the top-k outputs of a pretrained model (ZS in our experiments) is correct in a binary classification setup. Entities are then ranked based on the sum of the probabilities produced by the pretrained model and the score produced by the finetuned model. **Finetuning (FT):** where we finetune a model on the knowledge extraction task on the training set before evaluating on the test set.

### 3.5 Metrics

We report the results using the widely-used *Hit@k* metric computed as the percentage of queries for which the correct answer is ranked among the top *k* entities. We compute *Hit@k* for each relation type separately and report the macro average, following previous work.

## 4 Understanding Finetuning for Factual Knowledge Extraction

In this section, we design experiments that help better understand the effects of finetuning.

Table 2: Performances on the three datasets (bold indicates winner). FT offers substantial gains when train and test sets have similar frequency statistics, but the gain diminishes as the gap between the frequency statistics becomes more; eventually on *HighMismatch*, the negative side-effects outweighs the positive effects and finetuning becomes harmful overall.

| Strategy | *LowMismatch* | | | *MediumMismatch* | | | *HighMismatch* | | |
|---|---|---|---|---|---|---|---|---|---|
| | Hit@1 | Hit@3 | Hit@5 | Hit@1 | Hit@3 | Hit@5 | Hit@1 | Hit@3 | Hit@5 |
| ZS | 35.2 | 47.9 | 52.7 | 35.2 | 47.9 | 52.7 | 19.5 | 28.3 | 31.7 |
| FS | 47.0 | 56.1 | 57.2 | 42.8 | 50.9 | 52.5 | **27.0** | **33.4** | **34.9** |
| RR | 39.9 | 49.9 | 52.7 | 38.7 | 49.1 | 52.7 | 20.5 | 28.8 | 31.7 |
| FT | **51.9** | **68.4** | **73.9** | **43.6** | **57.8** | **63.2** | 18.0 | 27.4 | 32.4 |

## 4.1 Finetuning Performance Depends on Frequency Statistics

We first compare different model variants on the *LowMismatch* dataset. According to the results in Table 2, FT yields a significant boost compared to the other variants. This result is consistent with what has been already observed in existing literature (Fichtel et al., 2021). To understand where the improvement comes from, in Figure 2, we plot the improvement gained by the FT model over the ZS model on the *LowMismatch* dataset as a function of the entity coverage and Pearson correlation between the train and test sets. Specifically, for each relation type in the dataset, we measure the amount of entity coverage as well as the Pearson correlation between train and test sets, then group different relation types based on these metrics and average the relative improvements in each group. According to Figure 2, the improvements are mostly for those relation types that have a high entity coverage and high Pearson correlation.

Based on the above result, we hypothesize that part of the improvement obtained by the FT model on the *LowMismatch* dataset is due to biasing the pre-trained LM's prediction frequencies toward that of the answer set of the training data; since the train and test sets have similar frequency statistics, the frequency bias given to the model due to finetuning matches that of the test set and that results in some improvement. To verify this hypothesis, we next compare FT with the other variants on the *MediumMismatch* dataset where entity coverage is still high but Pearson correlation is low. The results in Table 2 show that while FT still gives a boost in performance, the gain is much lower compared to the *LowMismatch* case. As we will show in Section 4.2, the gap between the performance of the FT model on *LowMismatch* and *MediumMismatch* is mainly due to the difference in the frequency statistics in the train and test sets: finetuning biases the entity frequency of the LM predictions toward that of the training data but the new frequencies do not match with that of the test set on *MediumMismatch*.

Moreover, we compare FT with the other variants on the *HighMismatch* dataset where both entity coverage and Pearson correlation are minimal. The results in Table 2 show that the bias in prediction frequencies of the LM caused by finetuning in this case even outweigh the positive effect from *Task Learning* resulting in a model that actually harms the overall performance and produces inferior results compared to the ZS model.

To verify how the above observations are affected by the scale of the LM, we also compare the ZS and FT models on the three datasets when using the T5 1.1 Small model (60M parameters) instead of the T5 1.1 XXL (11B parameters). According to the results in Table 3, the small model shows a similar behaviour where FT provides a big boost on the *LowMismatch* dataset, but the amount of boost diminishes on *MediumMismatch* and finetuning becomes harmful on *HighMismatch*.

Table 3: Performance on the three datasets when using T5 1.1 small instead of XXL.

| Strategy | *LowMismatch* | | | *MediumMismatch* | | | *HighMismatch* | | |
|---|---|---|---|---|---|---|---|---|---|
| | Hit@1 | Hit@3 | Hit@5 | Hit@1 | Hit@3 | Hit@5 | Hit@1 | Hit@3 | Hit@5 |
| ZS | 18.4 | 23.2 | 24.4 | 18.4 | 23.2 | 24.4 | **12.6** | **15.5** | **16.4** |
| FT | **28.5** | **43.0** | **49.5** | **25.9** | **36.2** | **41.7** | 5.9 | 8.6 | 9.7 |

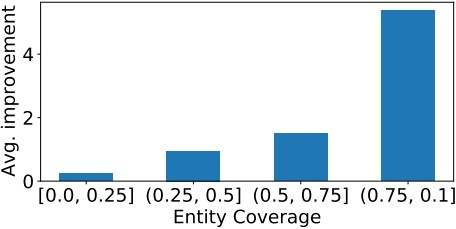 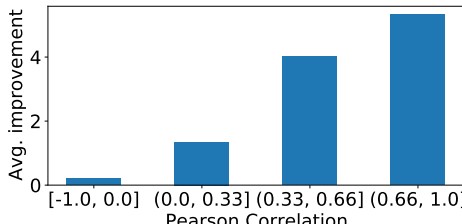

Figure 2: Macro average relative improvement of FT over ZS for different relation types in *LowMismatch* as a function of *entity coverage* and *Pearson correlation* for the train and test sets. The figures show that most of the improvement comes from the relations with a high entity coverage and Pearson correlation between train and test sets.

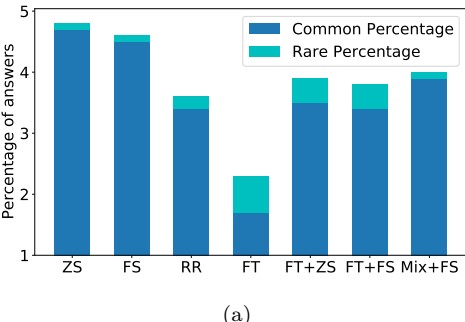

(a)

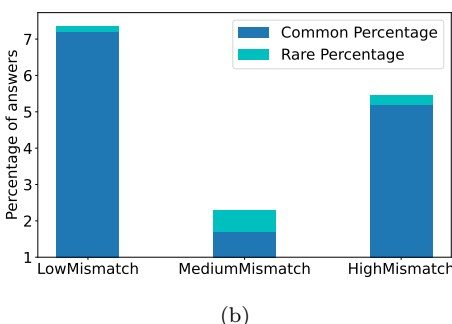

(b)

Figure 3: *Common (Rare) Percentage* corresponds to the percentage of test queries for which the model predicted an entity from the `Common` (`Rare`) set. (a) Results for different models on *MediumMismatch*, and (b) results for different datasets with the FT model. According to the results, after finetuning on the *MediumMismatch* dataset, the LM receives a frequency shock: it under-predicts the common entities and over-predicts the rare entities.

Despite the striking results obtained with finetuned LMs for factual knowledge extraction in previous work, the collective results in Table 2 show that (naively) finetuned LMs may not always be the best option for factual knowledge extraction and KG construction as the performance of these models depends heavily on the frequency statistics of the train and test sets.

## 4.2 *Frequency Shock* is a (Side-)Effect of Finetuning

We now design experiments that explain the behaviour observed for the FT model in Table 1 in terms of a *Frequency Shock* side-effect.

We selected a set of 10 cities that are expected to be commonly seen[3] as well as a set of 10 cities that appear as answers in LANKA but are expected to be rarely seen in a dataset[4]. We named the two sets `Common` and `Rare` respectively. We then measured the number of times the models generated an entity from `Common` and `Rare`.

In Figure 3(a), we report the percentages of `Common` and `Rare` for different models on the *MediumMismatch* dataset (We used *MediumMismatch* as its train set has a uniform distribution and makes *Frequency Shock* more candid). the ZS model predicts the `Common` entities frequently and the `Rare` entities infrequently (this is

---

[3]We selected the 10 cities from Cao et al. (2021) (Figure 2), namely {London, Paris, Tokyo, Boston, Rome, Chicago, Berlin, Montreal, Moscow, Milan}

[4]We do this by randomly selecting 10 cities from the LANKA answers that appear as an answer in LAMA less than 20 times, namely {Boise, Tirana, Myanmar, Hanover, Aberdeen, Chelsea, Kentucky, Oldham, Hastings, Parma}

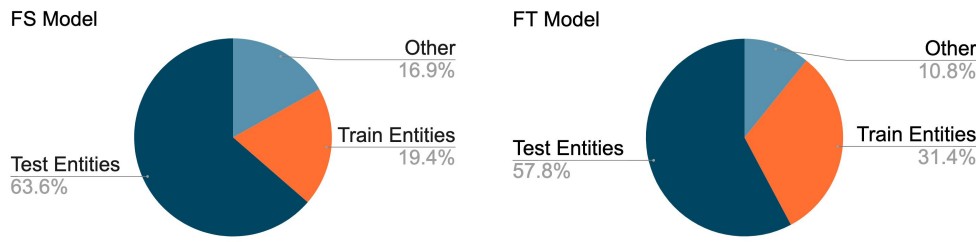

Figure 4: The percentages of overlap between the entities predicted by the models and those of the train and test sets for the *HighMismatch* dataset (entities in the train and test sets are disjoint).

in part due to the frequency of the entities in the test set and in part due to the prior of the language model). For the FS and RR models, the percentages for the two sets are not substantially different from the ZS model. However, for the FT model, due to the uniform distribution of the training set of *MediumMismatch*, the percentages for the two sets changes substantially: the number of predictions from `Common` entities drops by almost a third, and the one for `Rare` entities increases by 6x. This reveals that *Frequency Shock* is indeed a side-effect of finetuning as the difference between the frequency statistics of the training set of *MediumMismatch* and what the pre-trained model expects causes a shock to the model and makes it over-predict `Rare` entities and under-predict `Common` entities. The percentages for the FT model on the other datasets is reported in Figure 3(b). Unlike *MediumMismatch*, for *LowMismatch* `Common` entities are predicted frequently and `Rare` entities infrequently; for *HighMismatch* the frequency of `Common` entities goes down compared to *LowMismatch* because some of these entities do not appear in the train set.

We also measured the accuracy of the FT model when it produced a `Common` or `Rare` entity and compared it to ZS. The results are reported in Table 4. We observe that the accuracy for `Common` entities increases from 41.2% to 68.5% and for `Rare` entities decreases from 47.9% to 14.4%. This is because the *Frequency Shock* caused by finetuning leads the model to predict the `Common` entities only when it has high confidence in its prediction, but be less cautious about predicting the `Rare` entities.

Table 4: Accuracy of the models for the `Common` and `Rare` entity sets for the *MediumMismatch* dataset. Due to *Frequency Shock*, the FT model under-predicts the `Common` entities and over-predicts the `Rare` entities. As a result, when the FT model predicts a `Common` entity, there is a much higher chance of it being true compared to the other models, whereas when the FT model predicts a `Rare` entity, there is a much lower chance of it being true.

|  | Common Accuracy | Rare Accuracy |
|---|---|---|
| ZS | 41.2 | 47.9 |
| FS | 51.4 | 63.8 |
| FT | 68.5 | 14.4 |
| FT + FS | 57.7 | 29.5 |
| 1:15 + FS | 55.6 | 64.6 |

We also manually analyzed the outputs of the ZS and FT models for the "born in" relation (as a representative relation)[5] and grouped the predictions of each model into three classes: 1- the output is not a location, 2- the output is the correct location, and 3- the output is an incorrect location. We then compared the number of queries in the cross-product of the categories for the ZS and FT model. The results are presented in Table 5.

[5]We selected this relation because it is simple to verify the model's output types and subtypes.

Table 5: A comparison of the ZS and FT models for the "born in" relation. *NL*, *CL* and *IL* stand for *Not a Location*, *Correct Location*, and *Incorrect Location* respectively.

|  | FT | | | | | | | | |
|---|---|---|---|---|---|---|---|---|---|
|  | *LowMismatch* | | | *MediumMismatch* | | | *HighMismatch* | | |
| ZS | NL | CL | IL | NL | CL | IL | NL | CL | IL |
| NL | 0 | 11 | 105 | 0 | 3 | 113 | 0 | 0 | 165 |
| CL | 0 | 89 | 26 | 0 | 57 | 58 | 0 | 3 | 60 |
| IL | 0 | 115 | 598 | 0 | 62 | 651 | 0 | 2 | 838 |

Moreover, on *MediumMismatch*, out of the 58 queries for which the answer changed from a correct location to an incorrect location after finetuning, in 19% of those cases the correct entity was "London" – a commonly occurring city (note that only for 6% of the queries the correct answer is "London"); In another 33% of those cases the correct entity is one of "Paris", "Berlin", "Barcelona", "Vienna" and "Brooklyn", whereas only for 7% of the queries the answer is one of these cities. This is because the training set of *MediumMismatch* has a uniform distribution and finetuning on it leads to frequency shock where common entities (such as "London") are under-predicted.

***Frequency Shock* Causes *Range Shift*:**   As a specific case of *Frequency Shock*, we show that the range of the FT model changes mostly toward those entities seen as answers during finetuning. Toward this goal, we compare models in terms of the overlap between their predicted entities and those in the train and test sets of the *HighMismatch* dataset, where the train and test entities are mutually exclusive. According to the results in Figure 4, we observe that the FT model predicts the entities from the train set significantly more than the FS model (almost 62% relative increase). This shows a clear case of *Range Shift*. Moreover, in Table 5, out of the 60 queries on *HighMismatch* where ZS predicted the correct location and FT predicted an incorrect location, in 59 cases the top answer of the FT model was one of the entities from the training set answers, showing another clear (and perhaps more severe) case for *Range Shift*.

### 4.3  The positive effect of *Task Learning*

Similar to the existing literature (Fichtel et al., 2021), Table 5 provides multiple evidences showing *Task Learning* is a positive effect of finetuning. First, while ZS predicts non-location outputs (mostly years) for some queries, FT correctly learns to predict a location for the queries[6]. Secondly, for the 115 queries where ZS predicted an incorrect location but FT predicted a correct one on *LowMismatch*, in 90 cases the ZS model had generated a correct country as the top output, and the FT model learned to predict the correct city (which is the expected sub-type) instead of country. We observe a similar behaviour for 39/62 queries in *MediumMismatch*. The other cases where the prediction changed from incorrect location to correct location can be explained by better learning the semantics of the task as a result of finetuning.

## 5  Improving Finetuning

To avoid the side-effects identified in Section 4 and use finetuned LMs for factual knowledge extraction and KG construction, one may be tempted to create a training set that has a large coverage of various entities and that also has a high Pearson correlation with what is expected to be seen at the test time. We note, however, that entity coverage and Pearson correlation are somewhat at odds with each other. That is because if we add many queries to the training data whose answers are novel entities, it will cause the Pearson correlation to go down unless we also add a prohibitively large number of queries with common entities as answers to retain the proportions. Also, if we wish to keep the Pearson correlation high, many of the rare entities may not appear in the training set.

We aim to find solutions by changing the finetuning strategy. Given that LMs are pre-trained on large corpora of text (typically much larger than the finetuning dataset), we may expect the original entity distribution of the LM (corresponding to its prior distribution) to be more robust to situations with different frequency statistics. In this section, we exploit this insight to provide two strategies to remedy the negative effect of *Frequency Shock* and *Range Shift* in finetuning while still retaining the benefits of *Task Learning*.

### 5.1  Model Mixing

As we observed in the previous sections, the FT model has the advantage of better learning the task and as a result producing better results than the alternative models in situations such as the *LowMismatch* dataset. However, it also has the disadvantage of introducing a frequency bias that may lead to low performance on

---

[6]As an interesting side note, for the queries for which the ZS model outputs a different type than a location, even though the FT model learns to predict a location, it tends to predict a wrong location; future work can use this signal to predict when the LM does not know the answer to a question.

Table 6: Results for model mixing (bold indicates winner). The *best single model* corresponds to the model that gave the best result for each dataset (e.g., for T5 XXL FT is the best single model for *LowMismatch* and *MediumMismatch*, and FS for *HighMismatch*). *UB* stands for upper-bound.

| T5 | Model Mixing | *LowMismatch* | | | *MediumMismatch* | | | *HighMismatch* | | |
|---|---|---|---|---|---|---|---|---|---|---|
| | | Hit@1 | Hit@3 | Hit@5 | Hit@1 | Hit@3 | Hit@5 | Hit@1 | Hit@3 | Hit@5 |
| XXL | Best Single Model | 51.9 | 68.4 | 73.9 | 43.6 | 57.8 | 63.2 | **27.0** | 33.4 | 34.9 |
| | FT + ZS | 51.3 | 66.3 | 71.5 | 45.8 | 58.9 | 64.1 | 22.0 | 31.5 | 35.8 |
| | FT + FS | **53.5** | **69.0** | **74.2** | **46.9** | **60.9** | **66.1** | 26.2 | **34.4** | **38.2** |
| | FT + FS (UB) | 59.3 | 71.8 | 75.9 | 52.9 | 65.1 | 69.7 | 29.2 | 36.9 | 40.6 |
| Small | Best Single Model | 28.5 | 43.0 | 49.5 | 25.9 | 36.2 | 41.7 | 12.6 | 15.5 | 16.4 |
| | FT + ZS | **28.7** | **43.3** | **50.5** | **26.3** | **37.1** | **42.5** | **12.7** | **16.2** | **17.5** |

situations such as the *MediumMismatch* and *HighMismatch* datasets. We previously observed in Figure 3 and Table 4 that the FS model does not suffer from such a bias as it follows the predictive distribution of the pre-trained LM; we also observed that the FS model performs competitively with the FT model according to Table 2. Therefore, one possible way to alleviate the frequency bias of the FT model and improve its performance is to mix it with the FS model; since the predictive distribution of the FS model follows that of the pre-trained LM, mixing it with the predictive distribution of the FT model can help bring the FT distribution closer to the pre-trained model and alleviate the frequency bias. We experiment with a simple mixing approach where we average the scores produced by the FT model for each output with that of the other models; we leave more sophisticated combination strategies as future work.

**Model mixing alleviates side-effects:**  Figure 3 indicates the percentage of queries for which the FT+ZS and FT+FS models predicted one of the `Common` or `Rare` entities. One can see from the figure that, contrary to the FT model, the distributions for these models are much closer to that of the ZS and FS models. The correction effect is rather one-sided: while the `Common` entities are not under-predicted anymore, the `Rare` entities are still slightly over-predicted. This is also apparent from the accuracy of the FT+FS model on the `Common` and `Rare` sets in Table 4: the performance on the `Common` set becomes similar to the FS model as `Common` entities are not under-predicted anymore, but the performance on the `Rare` set is still much lower than the FS model as `Rare` entities are still being over-predicted.

**Model mixing leads to better performance:**  Table 6 reports the results for mode mixing on our datasets. When using the T5 XXL model, for FT+ZS even though the difference between the two models is quite large on *LowMismatch*, the model results are only slightly worse than the FT model itself. For the other two datasets, where the difference between the two models is much smaller, model mixing leads to substantial improvement. FT+FS offers higher performance than both individual models on *LowMismatch* and *MediumMismatch*. For *HighMismatch*, all numbers improve substantially with respect to FT; with respect to FS, however, Hit@1 goes slightly down whereas Hit@3 and Hit@5 improve. The same trend holds for the T5 Small model where mixing ZS and FT performs better than both ZS and FT in isolation. Note that for the *HighMismatch* dataset, even though the FT model performs poorly in isolation, mixing it with ZS still brings improvement for ZS.

While, in the past, model mixing has been shown to provide only marginal gains in different applications even when multiple models are being combined, the results in Table 6 show that a simple parameter-free combination of only two models provides large boosts of up to 7.6% on the *MediumMismatch* dataset. Furthermore, we have included in Table 6 an upper-bound result for FT+FS where we assume having access to an oracle that can tell if we should trust the FT model or the FS model for each query. The upper-bound result is significantly higher than each of the individual models, thus showing that there is a large subset of the data where one model produces the correct answer whereas the other model does not, hence indicating that the models work well on different subsets of the data and that more sophisticated combinations can potentially lead to more improvements. These results confirm that besides the previously studied benefits of

Table 7: Results for mixture finetuning with different mixture ratios (bold indicates winner). 1:0 corresponds to standard finetuning. Mixture training consistently provides a boost in performance, especially for larger mixture ratios. The benefits from mixture training and model mixing can be combined.

| T5 | Mixture | *LowMismatch* | | | *MediumMismatch* | | | *HighMismatch* | | |
|---|---|---|---|---|---|---|---|---|---|---|
| | | Hit@1 | Hit@3 | Hit@5 | Hit@1 | Hit@3 | Hit@5 | Hit@1 | Hit@3 | Hit@5 |
| XXL | 1:0 | 51.9 | **68.4** | 73.9 | 43.6 | 57.8 | 63.2 | 18.0 | 27.4 | 32.4 |
| | 1:1 | 51.8 | **68.4** | **74.0** | 45.6 | **61.0** | **66.9** | 18.2 | 27.3 | 32.3 |
| | 1:5 | 52.2 | 68.0 | 73.9 | **46.5** | 60.5 | 66.6 | 18.5 | 27.9 | 32.9 |
| | 1:15 | **52.7** | **68.4** | 73.9 | 45.6 | 60.2 | 66.1 | **19.5** | **28.8** | **33.4** |
| | 1:15 + FS | 53.4 | 69.2 | 74.4 | 47.2 | 63.0 | 68.6 | 26.7 | 35.1 | 39.0 |
| Small | 1:0 | 28.5 | 43.0 | 49.5 | 25.9 | 36.2 | 41.7 | 5.9 | 8.6 | 9.7 |
| | 1:15 | **31.6** | **47.9** | **54.2** | **26.2** | **36.8** | **42.9** | **10.5** | **13.0** | **15.5** |

model mixing (Naderi et al., 2021; Pranesh et al., 2020; Wang et al., 2022; Ormerod, 2022), it plays a much significant role for knowledge extraction from finetuned LMs by correcting the side-effects of finetuning.

## 5.2 Mixture Training Alleviates Side-Effects

Another solution to alleviating the frequency bias of the FT model is to fine-tune with a multi-task objective that mixes the factual knowledge extraction task with an auxiliary prediction task that ensures that common entities are observed frequently and rare entities infrequently. A natural candidate for the auxiliary task is the original pre-training task of the LM ("Masked Language Modeling"). We use the standard "text-to-text" formulation introduced in (Raffel et al., 2019) to implement the multi-task objective combining these two tasks. Let $\alpha : \beta$ represent the ratio between the number of queries from the main and auxillary tasks in each training batch. We set $\alpha = 1$ and finetune models with different values for $\beta$ for the three datasets, to see how mixture training with different ratios affects the model performance. A similar technique called "Mix-Review" was introduced in (He et al., 2021) to remedy the forgetting effect of finetuning LMs; a significant difference however is the use of a decay parameter in Mix-Review that reduces the proportion of the pre-training task in later epochs, our Mixture Training does not use this since it would negate the benefit of using the pre-training task to alleviate *Frequency Shock*.

**Mixture training leads to better results:** From the results in Table 7, we can see that mixture finetuning consistently provides improvements across the three datasets. This is true for both T5 XXL and T5 Small models. The amount of improvement is larger for the *HighMismatch* dataset where the statistics differ more. Interestingly, the biggest gains are seen for relatively high values of $\beta$. This may seem surprising, since it would be expected that the best performance would be obtained by a mixture dominated by the main task with the auxillary task as a regularizer. However, the higher proportion of the auxillary task is necessary to ensure that the entity distribution does not suffer a *Frequency Shock*, with enough of the main task mixed in that the resulting model acquires the *Task Learning* skill.

**Mixture training + model mixing alleviate side-effects:** We combine the mixture finetuned model with the FS model and see from Figure 3 that the resulting model does not under-predict `Common` entities and does not over-predict `Rare` entities. Also, from Table 4, we see that the resulting model does not produce a substantially higher accuracy on the `Common` set due to under-prediction, and does not produce a substantially lower accuracy on the `Rare` set due to over-prediction. This shows that the two solutions can be combined to effectively mitigate the side-effects. Finally, we see from Table 7 that the benefits from model mixing and mixture training can be combined to make yet better predictions.

# 6 Discussion, Future Directions, and Conclusion

**Summary of findings:** Language models (LMs), especially when finetuned, can be a great source of knowledge for constructing (or augmenting) knowledge graphs. However, finetuning may also exhibit negative effects for knowledge extraction that are important to understand and be aware of. We provide a high-level summary of our findings and their connections to existing work below.

- We introduced entity coverage and Pearson correlation as metrics to measure the difference in frequency statistics and showed that finetuning is effective mostly when these metrics are high between train and test sets.

- We identified *Frequency Shock* as a side-effect of finetuning which make finetuning harmful overall in certain situations with high frequency mismatch between train and test sets.

- We showed that compared to the previously identified forgetting effect, *Frequency Shock* provides a more nuanced explanation of the weaknesses of finetuned LMs for factual knowledge extraction.

- We showed that mixing a finetuned model with a fewshot (or zeroshot) model alleviates *Frequency Shock* and leads to substantial boosts, because the fewshot model follows the distribution of the pre-trained model.

- We showed that mixture training with the pre-training task alleviates *Frequency Shock* and leads to substantial boosts, because the pre-training task helps the LM keep its original frequency statistics.

We already discussed in Section 2 the works from the literature that closely relate to our work. In what follows, we discuss some higher-level connections and future directions.

**Connection to out-of-distribution generalization:** Classical machine learning settings assume train and test sets come from the same distribution. Recently, there has been much effort in tackling more realistic scenarios where test distributions differ from training distributions, known as out-of-distribution (OOD) generalization (see Shen et al. (2021) for a survey). While OOD generalization has been investigated in many applications, it has remained largely unexplored for factual knowledge extraction from LMs. This may be due to a lack of clarity on what a meaningful definition of OOD is for this task; since test queries are written using the same template as the training queries, traditional definitions are not straightforward to apply. Lewis et al. (2020) for example takes the extreme approach (in the context of Open-Domain Question-Answering) of defining OOD as queries whose answers have never been seen in training.

Using frequency statistics to measure the distance between train and test sets could be viewed as a novel formulation of OOD for factual knowledge extraction from LMs, and the negative side-effects discussed in Section 4 and the solutions considered in Section 5 are both relevant for robust solutions to OOD generalization. Note, however, that the *Frequency Shock* phenomenon goes beyond OOD generalization, e.g. in cases such as the example in Figure 1, the test query could still be an in-distribution query.

**Connection to domain adaptation:** In domain adaptation (Kouw & Loog, 2019), one has access to labeled examples $(x, y)$ from a source domain and unlabeled examples $(z)$ from a target domain. The goal is to train a model on the source data that works well on the target data. This is, e.g., done by weighing the training source examples based on their likelihood under the target distribution $p(z)$. In our problem, the finetuning data can be considered as the source data, but we do not have access to unlabeled examples $(z)$ from the target domain before deployment. However, given that LMs have been pre-trained on massive corpora of text, we might expect that $p(z)$ (i.e. the target distribution and frequency statistics) resembles that of the pre-trained LM and hence use the distribution of the pre-trained LM as an approximation. Our mixture training solution implicitly follows this intuition; future work can look into more direct ways of leveraging this connection.

**Connection to LMs knowing what they do not know:** An interesting research question that has recently received attention is to verify if LMs know what they do not know (Kadavath et al., 2022; Jiang et al., 2021). Indeed, factuality has been identified as the key shortcoming of recent powerful generative models such as ChatGPT that are fine-tuned on human feedback. One future direction is to see if there is a

benefit from *Frequency Shock* in terms of leveraging it for this problem. Specifically, we observed in Table 4 that finetuning on a dataset where common entities appear infrequently makes the model under-predict these entites, but the the accuracy of the predictions substantially increases. Therefore, one can measure whether a ZS (or FS) model knows what it does not know by comparing its predictions to an FT model that has been finetuned on a slightly different distribution.

**Connection to bias amplification:** Previous work has shown that datasets have several kinds of biases and models trained/finetuned on these models pick up those biases. As an example, Zhao et al. (2017) show that certain verbs are more associated with their stereotyped genders in some datasets, and a model trained/finetuned on such datasets amplifies that bias. To remedy this problem, one common practice is to create balanced datasets for finetuning where for each verb an equal number of examples from each gender appear in the dataset. Such an approach is, however, reminiscent of our *MediumMismatch* dataset and so *Frequency Shock* suggests that this might lead to a severe bias in the other direction, where the model mostly associates verbs with the non-stereotyped gender. We leave an empirical study of this effect as future work.

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

## A  Implementation Details

We train all models for 10 epochs on a $4x4$ v3 TPU. Note that in the case of mixture finetuning (Section 5.2), each epoch consists of more batch updates compared to single-task finetuning because a portion of the examples in each batch come from the pre-training task. We set the batch size to 128 and the learning rate to 0.0001. We measured the Hit@1 performance on the validation set after each epoch and selected the model parameters from the best performing epoch on the validation for evaluating on the test set.

For the few-shot experiments (FS model), for each relation type we selected 10 random examples from the training set of the dataset. To get an estimate of the standard deviation due to the choice of different examples, we repeated our experiment for the *LowMismatch* dataset 5 times each time selecting different examples and observed a standard deviation of 0.56.

Note that while for the three datasets we use in this paper the queries have been selected in such a way that the answer is mostly a single token according to the BERT vocabulary, those answers are already multi-token according to the T5 vocabulary and that allows us to test the multi-token prediction ability of the LMs. Previous work restrict the model prediction distribution to a predefined set of tokens and ignores any predicted outputs outside that set. We disregard that pre-defined set during training and validation to avoid unwanted artifacts introduced due to the use of that specific vocabulary set. However, we use that set for measuring performance on the test set so that the final results are in the same footing as those of the previously published work.

## B  Model Details

We used four models in this paper, zeroshot, fewshot, reranking, and finetuned. We provided a brief description in the main text. Here, we provide more detail on how each model works.

- **Zeroshot (ZS)**: For this model, the pre-trained model parameters remain fixed. For a test query such as *(Barack Obama, profession, ?)*, a template is used to turn it into a question in natural language form such as *"Barack Obama is a [MASK] by profession."*, where the *[MASK]* token is where the answer is expected to be. The question is fed to the LM for which the LM produces a distribution over sequences for the *[MASK]* token. We use the sequence with the highest probability as the answer.

- **Fewshot (FS):** This model is similar to the ZS model except that some example questions and answers are prepended at the beginning of the LM input. These examples are for the same relation

as that of the query to be answered. For *(Barack Obama, profession, ?)*, for example, the input of the LM may be as follows: *"Doug Rhodes is a keyboardist by profession. Greg Davidson is a referee by profession. Sandra Sarikakis is a cook by profession. Barack Obama is a [MASK] by profession."*. The fewshot examples are randomly selected from the training set and fixed for all examples. We tuned the number of fewshot examples on the validation set.

- **Reranking (RR)**: We first use the ZS model to predict a distribution over possible answers for the examples in the train and validation sets, and select the top-k ($k = 5$ in our experiments) answers. Then, we use the procedure outlined below to create a finetuning dataset based on the train set. For any question $q$ and answer $a$ in the top-k answers, we create a query similar to the ZS model, but where the *[MASK]* token is filled with $a$. For example, if for *(Barack Obama, profession, ?)* one of the top-k answers is *politician*, we create the query *"Barack Obama is a politician by profession."*. We assign a label of 1 to the query if $a$ is the correct answer to $q$ and 0 otherwise. We follow a similar procedure for the validation set. For the training set, if none of the answers in the top-k are correct, we also create another query where the *[MASK]* token is replaced with the correct entity and the label is 1. We then finetune an LM on this dataset for binary classification. During finetuning, we measure the performance of the LM on the validation set after each epoch and select the epoch that offers the highest *Hit@1* on the validation set. For testing, we get two scores for each example: 1- using the ZS model, 2- we obtain the top-k predictions of the ZS model and use the finetuned model to predict their correctness probabilities. We sum these two scores and select the answer with the highest total score as the final answer.

- **Finetuned (FT)**: Similar to the ZS model, we first convert queries into natural language form using templates. Then, we use the train set and finetune an LM with the masked language modeling objective, where the LM is finetuned to predict the correct answer for the *[MASK]* token given the query. For example, given query *"Barack Obama is a [MASK] by profession."*, the LM is finetuned to predict *"politician"* as output. During finetuning, we measure the performance of the LM on the validation set after each epoch and select the epoch that offers the highest *Hit@1* on the validation set. At the test time, we evaluate the LM similar to the ZS model but with the updated parameters.

