# OpenReview forum: "Understanding Finetuning for Factual Knowledge Extraction from Language Models"
_TMLR — Rejected by TMLR_

### Review · Reviewer_6iZL · 2023-02-12

**Summary Of Contributions:**

* The paper points out three effects of finetuning for factual knowledge extraction from language models: task learning, frequency shock, and range shift.
   - task learning: finetuning helps the model be more aware of the task of interest
   - frequency shock, and range shift: finetuning biases the model towards the entities in the training set, with their frequencies


* The paper conducted several experiments and show evidence of these effects
   - it introduces 2 novel metrics (entity coverage and Pearson correlation) to measure how the data distributions of the two datasets for this task differ
   - it creates novel ways to evaluate the three effects in the scenarios of  factual knowledge extraction from language models

* The paper proposes a simple way to mitigate the frequency shock and range shift problems, by mixing models

**Audience:**

Yes

**Broader Impact Concerns:**

There're no impact concerns

**Claims And Evidence:**

No

**Requested Changes:**

* I strongly recommend the authors to put the paper in the context of domain adaption.

* I strongly recommend the authors to describe the used approaches (zero-shot, few-shot, especially fine-tuning) with more details.

* Small changes:
   - Pearson correlation (sec 3.2): I think the two quantities in the two sums of the denominator should be squared.

**Strengths And Weaknesses:**

**Strengths**

* The paper examines the three effects of finetuning in a systematic way, including proposing new reasonable metrics (coverage and Pearson correlation),

* The paper is easy to follow.


**Weaknesses**

* In general, the paper doesn't properly address and put itself in the relevant context. In fact, what shown in the paper have been discussed very widely in **domain adaptation** (though not for  factual knowledge extraction from language models specifically). The crucial concept here is *domain shift* (i.e. the discrepancy between the domain of the train data, and the domain of the test data) which is quantified by "distribution shift" (see [1] for an overview).

* Finetuning is the crucial concept in the paper, but it is poorly described. Most importantly, how was a language model finetuned in the experiments? If there's a proper early stopping on a dev set that has similar data distributions like the test set, we shouldn't observe any significant drop over zero-shot.

* Section 5.3 "A Knowledge Extraction Recipe" doesn't seem realistic as it ignores the reality of real test data distribution. Hence, the "recipe" here is hardly seen as a generic advice, rather it is based solely on the way the experiments were conducted.


[1] Kouw, Wouter M., and Marco Loog. "A review of domain adaptation without target labels." IEEE transactions on pattern analysis and machine intelligence 43.3 (2019): 766-785.





------------------
I would like to thank the authors for detailed feedback with patience. For the latest version of the submission, I would like to strikethrough the last two weaknesses mentioned in the review, because:

   1. the paper presents in details the finetuning method in Appendix A,
   2. section 5.3 is removed.

I however would like to argue that the evidence for *frequency shock* is not clear as the paper does not proof whether this phenomenon happened by chance or not. This is because
   1. The paper shows frequency shock only on one dataset (MediumMismatch). Is it also true for other datasets?
   2. Even the paper does not show clear evidence for MediumMismatch. Only via the following responses (not in the paper), it turns out that the training set contains more common entities than rare ones (435 vs 355). But the drop of common percentage in fig 3a could be due to different reasons rather than frequency shock; for instance, due to the context given to the model (e.g. the query template, asked relations).

Briefly, although I appreciate the great effort of the authors in replying my review, I still keep my criticism that the paper is not strongly backed by the shown evidence. However, I accept the authors' argument that the paper could attract some attention from the corresponding community.

---

> ### Author Response · Authors · 2023-02-21
> **Author Response**
>
> We thank the reviewer for constructive feedback. We hope that our revised version and the following responses address your comments satisfactorily, and we would be happy to discuss and address any other concerns.
>
> > The paper proposes a simple way to mitigate the frequency shock and range shift problems, by mixing models
>
> We wish to clarify that the recipe we propose has 2 parts: the model mixing mentioned by the reviewer and “mixture training” (sec 5.2) i.e. multi-task training with the pre-training task. In sec 5.2 we have added some further explanations of the utility of mixture training.
>
> > In general, the paper doesn't properly address and put itself in the relevant context. In fact, what shown in the paper have been discussed very widely in domain adaptation (though not for factual knowledge extraction from language models specifically). The crucial concept here is domain shift (i.e. the discrepancy between the domain of the train data, and the domain of the test data) which is quantified by "distribution shift" (see [1] for an overview).
>
> We would like to emphasize that there are key differences between our work and the domain adaptation literature. Most notably, in domain adaptation one aims to adapt a model trained on a domain/distribution A to a target domain/distribution B; in our case, however, we aim to retain as much as possible the frequency statistics learned from domain A (i.e. the pre-training task) and avoid adapting to the target domain B (i.e. the finetuning dataset). Moreover, while the classic distribution shift literature considers two distributions (that of train and test), here we are dealing with three distributions: pre-training distribution, finetuning distribution, and test distribution. We also discussed in Section 6 of our revision the difference between our work and OOD generalization, which is another closely related but fundamentally different problem.
>
> > Finetuning is the crucial concept in the paper, but it is poorly described. Most importantly, how was a language model finetuned in the experiments? If there's a proper early stopping on a dev set that has similar data distributions like the test set, we shouldn't observe any significant drop over zero-shot.
>
> We explained all models in detail in the appendix (highlighted in blue). We assume the validation set follows a similar distribution as the train set (and not as the test set), and we select the best epoch on the validation set. This is a common and practically reasonable assumption, because one may not a-priori know what distribution the test set will follow. (Note that the dataset creation procedure was already explained in detail in 3.3 and model selection based on validation set was explained in detail in Appendix A).
>
> > Section 5.3 "A Knowledge Extraction Recipe" doesn't seem realistic as it ignores the reality of real test data distribution. Hence, the "recipe" here is hardly seen as a generic advice, rather it is based solely on the way the experiments were conducted.
>
> First, we would like to emphasize two things: 1) the datasets we used are generic datasets built by the knowledge extraction community to be representative of the task, so we expect the experimental results to be widely applicable, 2- Our top recommendation (1:15 + FS) performs best on every individual dataset (including the LowMismatch dataset which is the most frequently used dataset/setting in the literature).
>
> We agree with the reviewer that the best single model is contextual on the amount of frequency mismatch we expect. In our revision, we changed the language and added further explanation to highlight this (see the highlighted text in blue).
>
> > Pearson correlation (sec 3.2): I think the two quantities in the two sums of the denominator should be squared.
>
> Thanks, we fixed it in the submitted revision.

---

> > ### Comment · Reviewer_6iZL · 2023-02-22
> > **reply**
> >
> > I would like to thank the authors for the response.
> >
> > First of all, I would like to address the relation of this work and domain adaption literature. The literature presents a wide range of approaches (and frameworks) to the problem of the mismatch between the data distribution of a train set, and the data distribution of a test set. And I believe this is the problem that the paper tackles: both frequency shock and range shift can be framed as prior shift p(label) (and perhaps posterior shift p(label | input)).
> >
> > The authors argue that
> >
> >  > Most notably, in domain adaptation one aims to adapt a model trained on a domain/distribution A to a target domain/distribution B; in our case, however, we aim to retain as much as possible the frequency statistics learned from domain A (i.e. the pre-training task) and avoid adapting to the target domain B (i.e. the finetuning dataset)
> >
> > I want to correct it that the target domain B however, in domain adaption, is the domain of the test set, not the finetuning dataset.
> >
> > I don't see why  "to retain as much as possible the frequency statistics learned from domain A (i.e. the pre-training task)". When saying that, I believe the authors assume that the distribution gap between the distribution of pre-training data and that of test data is not large enough for getting negative impacts. If not that, I don't see how one can void negative impacts. For instance, if we pre-train a model on a very large text from fantasy novels, how could this help the model answers math questions? And in this case, do we really want to retain the knowledge learned from fantasy novels?
> >
> > Re section 5.3, "A Knowledge Extraction Recipe", I don't see any universal evidence for the recommendation. In fact, I'm questioning how the the ratio is chosen without knowing anything about test sets (e.g. the example with fantasy novels above). As the ratio is a hyper-param, we need to tune it. Here, the authors tune it on the created test sets. But what if new test sets are very different from the test sets used in the paper?

---

> > > ### Author Response · Authors · 2023-02-24
> > > **Author response**
> > >
> > > We had previously submitted our revision to the supplementary material by mistake. We fixed this and you should be able to see the revision now. We apologize for any confusion this might have caused.
> > >
> > > > First of all, I would like to address the relation of this work and domain adaption literature. The literature presents a wide range of approaches (and frameworks) to the problem of the mismatch between the data distribution of a train set, and the data distribution of a test set. And I believe this is the problem that the paper tackles.
> > >
> > > For our problem, the predictive distribution p(label | input) is not expected to change, since, after all, the answer to a real-world factual query cannot change with the statistics of the dataset. P(features) do change on fine-tuning, and hence so does P(label).
> > > We believe that our problem is more similar to “domain/OOD generalization” as opposed to “domain adaptation” for the following reason:
> > > The definition of domain adaptation assumes the existence of labeled examples (x, y) from a source domain and the existence of unlabeled examples (z) from a target domain (this is true both in the definition provided in the paper shared by the reviewer as well as other prominent works such as [1] – definition 1). In our problem, we do not have access to unlabeled examples (z) from the target domain before deployment.
> > >
> > > Not having access to unlabeled examples (z) makes our problem an instance of the “domain generalization” problem (see definition 2 of [1]), and since we expect P(features) and P(label) to change in deployment due to the nature of the problem, this makes our problem more more resemble “OOD generalization”.
> > >
> > > Despite the above distinction, we agree with the reviewer that some techniques from the domain adaptation literature may still apply to our setting. For example, even though we do not have access to unlabeled examples (z) from the target domain, there might still be a way of estimating P(z) based on the distribution of the pre-trained LM. Our mixture training solution follows a similar intuition but future work can indeed look into more direct ways of leveraging this connection.
> > >
> > > We have framed our problem as a form of OOD generalization in  Sec 6. Based on the reviewer’s feedback, we can also highlight it more centrally in the introduction (where we have only referred to it once). Moreover, if the reviewer agrees with the above connection between our work and domain adaptation, we are happy to add it to the paper.
> > >
> > > [1] Scatter Component Analysis: A Unified Framework for Domain Adaptation and Domain Generalization
> > >
> > > > both frequency shock and range shift can be framed as prior shift p(label) (and perhaps posterior shift p(label | input))
> > >
> > > We would like to re-emphasize that frequency shock goes beyond domain generalization: Frequency shock is more related to the concept of bias amplification (pointed out by RnC5) showing that the model substantially amplifies the frequency mismatches between the pre-train and finetune sets.
> > >
> > > > I don't see why "to retain as much as possible the frequency statistics learned from domain A (i.e. the pre-training task)". When saying that, I believe the authors assume that the distribution gap between the distribution of pre-training data and that of test data is not large enough for getting negative impacts.
> > >
> > > In our mixture training proposal, we are assuming a LM finetuned on a general corpus of texts such as C4 rather than a specialized domain (we have clarified this in the latest revision).  The assumption (which is verified by our experiments) is that this leads to a more general (i.e. not specialized) distribution of entities learned by the fine-tuned model.
> > >
> > > > For instance, if we pre-train a model on a very large text from fantasy novels, how could this help the model answers math questions?
> > >
> > > Our problem is factual knowledge extraction (for the purpose of knowledge base construction) and we indeed assume that LM has been pre-trained on a corpus of relevant knowledge, the finetuning is on task-specific data, and the test is also on the task-specific data. The problem formulation where pre-training, finetuning, and test are from different task domains is a much more challenging problem (as already shown by the reference Wallat et al. in the paper) and is beyond the scope of our work.

---

> > > > ### Author Response · Authors · 2023-02-24
> > > > **Author response continue**
> > > >
> > > > Continuing the response.
> > > >
> > > > > Re section 5.3, "A Knowledge Extraction Recipe", I don't see any universal evidence for the recommendation. In fact, I'm questioning how the the ratio is chosen without knowing anything about test sets (e.g. the example with fantasy novels above). As the ratio is a hyper-param, we need to tune it. Here, the authors tune it on the created test sets. But what if new test sets are very different from the test sets used in the paper?
> > > >
> > > > The reason we consider this a good default recommendation is that the 1:15+FS model is strictly better than the baseline single models on all combinations of datasets. So whatever the characteristics of the deployment distribution (which being an OOD setting we have little access to), we can have confidence that our proposed models are better choices for.
> > > > Note that in our revision, we have toned down the recipe by emphasizing the connection to the problem settings we considered in the paper.
> > > >
> > > > > Re audience:
> > > >
> > > > We noticed that the reviewer responded “no” to the audience question. We would like to emphasize that we study a highly practical problem in our paper: if one wants to extract a knowledge base from an LM, should they finetune on task-specific data or not? With knowledge extraction from LMs becoming a prominent approach to knowledge base construction and hallucination being a central challenge of the modern pre-trained LMs such as ChatGPT, we believe insights about this question will be of immense practical use. We appreciate it if the reviewer can provide more details on their judgement regarding the audience of our work.
> > > >
> > > > > Re claim and evidence match:
> > > >
> > > > We noticed that the reviewer responded “no” to the claim and evidence match question. With the “recipe” section toned down in the revision, we appreciate it if the reviewer provides some clarity on the claim(s) for which the reviewer believes we did not provide enough evidence for.

---

> > > > > ### Author Response · Authors · 2023-02-27
> > > > > **Discussion Phase Ending Soon**
> > > > >
> > > > > Dear Reviewer,
> > > > >
> > > > > We deeply appreciate your efforts and time in reviewing our paper and your valuable feedback so far. We have carefully responded to your comments. Before the discussion phase ends (which is soon), we would like to have the opportunity to address any other concerns you may have and revise our manuscript accordingly. We appreciate it if you could please read our responses and give us some feedback.
> > > > >
> > > > > Best,
> > > > >
> > > > > Authors

---

> > > > > > ### Comment · Reviewer_6iZL · 2023-02-28
> > > > > > **reply**
> > > > > >
> > > > > > Many thanks to the authors for the detailed response, which shed light to several unclear points.
> > > > > >
> > > > > > First of all, I would like to clarify my point of connecting the work with domain adaptation. We should notice that, by using frequency statistics to measure the distribution differences between train and test sets, and later coming up with mixing strategies, the work implicitly tries to close the gap between train and test set statistics, with some knowledge about the distribution of the target domain. For example, in sec 5.2 "... is to fine-tune with a multi-task objective that mixes the factual knowledge extraction task with an auxiliary prediction task that ensures that common entities are observed frequently and rare entities infrequently". This idea implies that in testing time, common entities are supposed to be observed more frequently than rare entities. Otherwise, there shouldn't be any reason for using an auxiliary task.
> > > > > >
> > > > > > The response mentions *bias amplification* "showing that the model substantially amplifies the frequency mismatches between the pre-train and finetune sets". Is this surprising? I hardly think so. In terms of Bayesian, the pre-trained model provides prior, and finetuning is updating the prior belief with observation for posterior. When more and more observations come (e.g. training set), the gap between prior and posterior is **obviously** getting larger and larger if observations are different from prior belief.
> > > > > >
> > > > > > Continuing the discussion on why "to retain as much as possible the frequency statistics learned from domain A (i.e. the pre-training task)", I would like to use another example where similar domains can be different. A LM trained on the whole Internet's texts before 2021 will fill "Donald Trump" to "[mask] is president of the USA" because obviously "Donald Trump" was the most common entities for that relation. However, this answer is no longer true. Therefore, we shouldn't always "retain as much as possible the frequency statistics learned from domain A". The question we should answer here is which frequency statistics we want to "retain as much as possible", and which statistics we expect the finetuning dataset to capture.
> > > > > >
> > > > > > Re discussion on sec 5.3, I would remind the authors that the recipe *1:15+FS* is concluded based on the results on the test sets. Meaning that the work uses the test sets to **tune the hyper-params** (e.g. 1:15). Furthermore, that "we expect knowledge extraction in real applications to involve a combination of the three datasets studied in this work" is a too strong claim. Unless the authors could prove that, this is just a speculation (e.g. randomly select datasets in the real world and examine whether they are a combination of the three datasets used in the paper). Last but not least, this recipe is model-specific.
> > > > > >
> > > > > > Re decision on "audience"
> > > > > > > ... we study a highly practical problem in our paper: if one wants to extract a knowledge base from an LM, should they finetune on task-specific data or not? ... immense practical use.
> > > > > >
> > > > > > I would like to keep my decision *no* because the question is not the matter here. This is because no pre-trained LMs has reached a sufficiently high performance for the task of KB extraction (e.g. the paper shows ZS can't get higher than 19% hit@1), and thus finetuning them is a must. Besides, a pretrained LM (as well as a KB) can't cover every corner of knowledge.
> > > > > >
> > > > > > Re claim and evidence. The main reason I keep my decision *no* is section 5.3. I would be happy to change my decision to "partial" if that's an option.

---

> > > > > > > ### Author Response · Authors · 2023-02-28
> > > > > > > **Author Response**
> > > > > > >
> > > > > > > We thank the reviewer for their response and clarifications.
> > > > > > >
> > > > > > > > First of all, I would like to clarify my point of connecting the work with domain adaptation. We should notice that, by using frequency statistics to measure the distribution differences between train and test sets, and later coming up with mixing strategies, the work implicitly tries to close the gap between train and test set statistics, with some knowledge about the distribution of the target domain. For example, in sec 5.2 "... is to fine-tune with a multi-task objective that mixes the factual knowledge extraction task with an auxiliary prediction task that ensures that common entities are observed frequently and rare entities infrequently".
> > > > > > >
> > > > > > > This is indeed what we tried to explain in our previous response. We are glad to see that we are in agreement with the reviewer. In our latest revision, we added a subsection in the discussion section that describes the connection. Thanks for bringing this connection to our attention.
> > > > > > >
> > > > > > > > The response mentions bias amplification "showing that the model substantially amplifies the frequency mismatches between the pre-train and finetune sets". Is this surprising? I hardly think so. In terms of Bayesian, the pre-trained model provides prior, and finetuning is updating the prior belief with observation for posterior.
> > > > > > >
> > > > > > > Let D1 be the frequency statistics of the pre-trained model, D2 be that of the finetuning data, and D3 be that of the finetuned model. If D3 was somewhere between D1 and D2, then we would agree with the reviewer that this would not be surprising and totally expected. However, the surprising part is in the amplification: entities that are frequent in D1 and less frequent in D2 become extremely infrequent in D3 and entities that are infrequent in D1 and more frequent in D2 become extremely frequent in D3 (see the distributions in Figure 3(a) and Table 5).
> > > > > > >
> > > > > > > We would like to also point out that the objective of TMLR is the claim and evidence match as opposed to the degree of surprising-ness.
> > > > > > >
> > > > > > > > Continuing the discussion on why "to retain as much as possible the frequency statistics learned from domain A (i.e. the pre-training task)", I would like to use another example where similar domains can be different. A LM trained on the whole Internet's texts before 2021 will fill "Donald Trump" to "[mask] is president of the USA" because obviously "Donald Trump" was the most common entities for that relation.
> > > > > > >
> > > > > > > Temporal obsolescence is indeed a big issue with current LMs and there is a large body of literature studying ways to resolve this issue with separate modeling techniques (see, e.g., [1]). Given that, we have kept temporal obsolescence out of scope for this paper.
> > > > > > >
> > > > > > > [1] Time-Aware Language Models as Temporal Knowledge Bases
> > > > > > >
> > > > > > > > I would like to keep my decision no because the question is not the matter here. This is because no pre-trained LMs has reached a sufficiently high performance for the task of KB extraction (e.g. the paper shows ZS can't get higher than 19% hit@1), and thus finetuning them is a must. Besides, a pretrained LM (as well as a KB) can't cover every corner of knowledge.
> > > > > > >
> > > > > > > We respectfully (strongly) disagree with the reviewer for the following reasons:
> > > > > > > *  There have already been several successful attempts in creating and/or augmenting knowledge bases from LMs (see many references in the Related Work section, KG Construction paragraph).
> > > > > > > *  The results in this paper are based on (publically available, reproducible) T5. There currently exists much bigger LMs with much higher factual accuracy, making KG construction from LMs a widely viable solution.
> > > > > > > *  The first paper in this area (the LAMA probe paper) has 1152 citations at the time of writing this comment, despite being only 3 years old. This hints at the size of the audience of this problem.
> > > > > > > *  In “the paper shows ZS can't get higher than 19% hit@1”, could you clarify which result you are referring to? Even when we took the macro average in Table 8 (removed in latest revision), the ZS model has 30.0% Hit@1. In other cases, the performance is even higher.
> > > > > > > *  “thus finetuning them is a must”: Agreed, in fact we have not concluded in our paper that we should NOT finetune LMs. We present an approach that mixture finetunes them and combines with a FS model.
> > > > > > > *  “Besides, a pretrained LM (as well as a KB) can't cover every corner of knowledge”: For many applications, we do not need exhaustive coverage of all areas. Similar to traditional approaches to KB construction, LMs can be used in a human-in-the-loop system to enable covering corner cases.
> > > > > > >
> > > > > > > > Re claim and evidence. The main reason I keep my decision no is section 5.3. I would be happy to change my decision to "partial" if that's an option.
> > > > > > >
> > > > > > > Upon the reviewers comments and suggestions, we decided to remove section 5.3 in our latest revision.

---

> > > > > > > > ### Comment · Reviewer_6iZL · 2023-02-28
> > > > > > > > **Reply**
> > > > > > > >
> > > > > > > > I thank the authors for the response.
> > > > > > > >
> > > > > > > > By now I would like to focus on *frequency shock*, which I want to see the evidence for it. The authors argue that
> > > > > > > >
> > > > > > > > > Let D1 be the frequency statistics of the pre-trained model, D2 be that of the finetuning data, and D3 be that of the finetuned model. If D3 was somewhere between D1 and D2, then we would agree with the reviewer that this would not be surprising and totally expected. However, the surprising part is in the amplification: entities that are frequent in D1 and less frequent in D2 become extremely infrequent in D3 and entities that are infrequent in D1 and more frequent in D2 become extremely frequent in D3 (see the distributions in Figure 3(a) and Table 5).
> > > > > > > >
> > > > > > > > I can see that, in Fig 3a, FT yields similar percentages for both Common and Rare. But isn't that due to the impact of the uniform distribution of the training set (LANKA)? If the percentage of Common were significantly lower than that of Rare, this would be the evidence of *amplification* like the argument of the authors. Also in Tab 5, what I can see is that frequent entities (e.g. London) is predicted less frequently, but I can't see evidence that those frequent entities were suppressed so that they were significantly dominated by infrequent entities.

---

> > > > > > > > > ### Author Response · Authors · 2023-03-01
> > > > > > > > > **Author Response**
> > > > > > > > >
> > > > > > > > > We would like to clarify that in LANKA, the distribution is uniform for each relation, but not across the entire dataset. That is, if e1 and e2 appear as answers for a relation r1, they appear an equal number of times. But e1 may also appear as answer for another relation r2, whereas e2 does not. Overall, our "Common" entities appear 435 times as answers in LANKA and "Rare" entities appear 355 times. (We agree with the reviewer that the term "amplification" used in our previous response may not be the right term in our case).

---

### Review · Reviewer_qrcB · 2023-02-13

**Summary Of Contributions:**

This paper studied the effect of fine-tuning pre-trained language models for factual knowledge extraction. The paper offered insights into the well-known forgetting effect by identifying a more detailed theory called "Frequency shock" and "Range shift." The authors proposed two methods to compute the mismatch between training and testing datasets. The experiments analyzed performances of different knowledge extraction methods (including fine-tuning) using different degrees of mismatch (low, medium, high).

The results support the claim that Frequency shock and Range shift are the side effects of fine-tuning where there is a mismatch between training and testing datasets. We can see the frequencies of the rare and common entities change after fine-tuning. The performance also drops significantly when the mismatch is high.

In addition, the authors also offered methods to mitigate the problem by mixing fine-tuned model's prediction and the few-shot prediction or mixing the mask language modeling sample during the fine-tuning. The experiments confirm that both are beneficial.

**Audience:**

Yes

**Broader Impact Concerns:**

Not related. This work does not carry a significant risk of harm.

**Claims And Evidence:**

Yes

**Requested Changes:**

### Changes

1. The "Frequency shock" is more dominant in both abstract and the main text. I think the authors could remove "Range shift" from the paper.
2. The author should briefly discuss the previous work on the effect of fine-tuning, such as Wallat et al., 2021.
3. The author should provide a more inclusive analysis of the frequency shock relationship with performance gain/loss (similar to Figures 2 and 3).
4. The author should provide a separate discussion to summarize the primary findings and relate the findings to other existing theories/methods, and discuss secondary findings and future work.

### Minor changes

1. Page 8, "of of" -> "of"
2. Citation formats were often incorrect in the text (e.g., use parenthesis form where it should not).

**Strengths And Weaknesses:**

### Strengths

1. The paper provided many experiment results to support its claims. The experiments with different degrees of mismatch and common vs. rare entities were well-designed to understand the underlying fine-tuning problem.
2. The paper provided more insights into the forgetting effect and OOD problems. It showed that the forgetting effect (or part) was from the frequency shock. In addition, it proposed a method to characterize the OOD in knowledge extraction tasks formally.

### Weaknesses

1. The proposed theories of frequency shock might be novel findings, but range shift is very similar to the forgetting effect. In addition, it was odd that the authors only showed Figure 2 with "Low mismatch" and Figure 3 with "Medium mismatch."
2. I think the proposed solution is still theoretically and empirically weak. Since the paper argues the root cause of the "frequency shock" effect, it does not elaborate clearly on why mixing would mitigate the problem, and it does not confirm that the proposed solution mitigates the root cause.
3. The organization of the paper was quite confusing. The paper mixed explaining the primary method/results with a discussion of relevant concepts, but not related to the main message. For example, Section 3.6 did not offer additional information on the method or experiment setup. Several paragraphs in the result sections also mentioned other directions.

---

> ### Author Response · Authors · 2023-02-21
> **Author Response**
>
> We thank the reviewer for constructive feedback. We hope that our revised version and the following responses address your comments satisfactorily, and we would be happy to discuss and address any other concerns.
>
> > it was odd that the authors only showed Figure 2 with "Low mismatch" and Figure 3 with "Medium mismatch."
>
> In the MediumMismatch dataset, the frequency statistics for the entities is the same, so drawing Figure 2 for this dataset may not be very meaningful. However, upon your suggestion, in our revision we expanded Figure 3 with results for the other two datasets.
>
> > Since the paper argues the root cause of the "frequency shock" effect, it does not elaborate clearly on why mixing would mitigate the problem, and it does not confirm that the proposed solution mitigates the root cause.
>
> Thanks to your comment, we noticed that we did not explain clearly how we expect the two proposed solutions to solve the frequency shock problem. We added explanations for both cases in the submitted revision (highlighted in blue).
> For confirming that the proposed solution mitigates the root cause, we would like to refer to Figure 3 and Table 4 of the paper:
> *  Figure 3 shows that both solutions help alleviate the frequency shock as the frequency predictions of the Common and Rare entities resembles that of the ZS and FS models.
> *  According to Table 4, the alleviation of frequency shock in the mixing models results in a more reasonable accuracy comparison across common and rare entities.
>
> > The "Frequency shock" is more dominant in both abstract and the main text. I think the authors could remove "Range shift" from the paper.
>
> We agree that frequency shock is the main highlight of the paper and we had already mentioned that range shift is a specific case of frequency shock. Upon your suggestion, in our revision we relegated the range shift effect to a sub-effect of frequency shock and only discussed it briefly in Section 4.2.
>
> > The author should briefly discuss the previous work on the effect of fine-tuning, such as Wallat et al., 2021.
>
> We added the main finding from Wallat et al. that relates to our work in the introduction (highlighted in blue). There are other findings in Wallat et al. with respect to the forgetting effect that do not directly relate to our work (e.g., they find that if the finetuning task is different from the knowledge extraction task, then forgetting is more severe; in our case, the finetuning task is the same as the original task).
>
> > The author should provide a more inclusive analysis of the frequency shock relationship with performance gain/loss (similar to Figures 2 and 3).
>
> This is indeed what we do in Table 4. We show that when the frequency shock is high (e.g., for the FT model), the model tends to have a high accuracy on common entities (but under-predict them according to Figure 3) and have a low accuracy on rare entities (but over-predict them). The two solutions we provided not only balance the under-/over-predictions but also keep the accuracy balanced between the common and rare entities, according to Table 4.
>
> > The organization of the paper was quite confusing. The paper mixed explaining the primary method/results with a discussion of relevant concepts, but not related to the main message. For example, Section 3.6 did not offer additional information on the method or experiment setup. Several paragraphs in the result sections also mentioned other directions.
> > The author should provide a separate discussion to summarize the primary findings and relate the findings to other existing theories/methods, and discuss secondary findings and future work.
>
> Thanks for the suggestion. We added a discussion section and 1) provided a summary of the primary/secondary findings and future work, and 2) moved to this section those parts of the paper that interrupted the main flow and caused confusions.
>
> > Minor changes
>
> Thanks, we fixed the typos and the citation format issues in the submitted revision.

---

> > ### Comment · Reviewer_qrcB · 2023-02-23
> > **Thanks**
> >
> > Thank you for responding to my comments and revising the manuscript according to the reviewers' suggestions.
> >
> > > For confirming that the proposed solution mitigates the root cause, we would like to refer to Figure 3 and Table 4 of the paper.
> >
> > Thank you for clearing that up. I believe this is a good confirmation of the proposed solution.
> >
> > For the inclusive analysis, I originally intended to ask about the other mismatch datasets. With the additional Figure 3(b) and its discussion, that should be sufficient.
> >
> > PS. For some reason, I still saw the same paper in the revision.

---

> > > ### Author Response · Authors · 2023-02-24
> > > **Author response**
> > >
> > > We had previously submitted our revision to the supplementary material by mistake. We fixed this and you should be able to see the revision now. We apologize for any confusion this might have caused.
> > >
> > > We are glad that we addressed your concerns and we are happy to discuss/address any other concerns you may have.

---

### Review · Reviewer_RnC5 · 2023-02-15

**Summary Of Contributions:**

This paper studies the effects of fine-tuning on factual knowledge prediction (e.g., "X was born in ___"). One of the headline results is an effect called frequency shock: rare entities observed frequently in the training set become very overrepresented, and common entities observed infrequently become very underrepresented. Furthermore, "Range Shift" is also observed, where the model mostly predicts answers observed during fine-tuning and loses some of its zero-shot capabilities. Three different train/test setups are derived from existing datasets to assess this.

Results show differing trends between LowMismatch, MediumMismatch, and HighMismatch setting. In particular, fine-tuning (FT) does better in the former two, but few-shot (FS) does best on the last. The gains on fine-tuning seem to largely be on common entities between train and test.

The paper then explores Frequency Shock and Range Shift in more detail. A fine-tuned model overpredicts rare entities and underpredicts common entities compared to a few-shot model.  FT also does much better on common entities compared to rare entities, whereas ZS/FS are comparable (or even better on rare). However, fine-tuning does help the model learn to predict the right type of entity for the task.

Finally, "model mixing" (combining the scores) of FT and ZS/FS leads to improvements, leveraging the complementary strengths of these two approaches. Some suggestions are given about how to apply these models going forward.

This paper's contributions/claims are:

1. Identifying Frequency Shock and Range Shift

2. Creating new datasets to study these

3. Identifying solutions and proposing a recipe for knowledge extraction

**Audience:**

Yes

**Claims And Evidence:**

Yes

**Requested Changes:**

Overall, I feel like this paper accomplishes what it sets out to do. I believe it is well-suited for TMLR, as it presents a contribution (albeit a slightly narrow one) and supports that contribution convincingly.

**Strengths And Weaknesses:**

STRENGTHS

First of all, this paper is very clearly written. It describes the backdrop for the knowledge extraction it's doing in good detail and has a thorough accounting of prior work.

This paper presents a clear hypothesis about Frequency Shock and Range Shift. In my view, these are reasonably well-supported. I say "reasonably" because I think the overall setting is a bit narrow: it's focused on a particular relation extraction setting with a particular set of datasets and entities, restricted to English. However, I think the overall story is plausible enough that I am willing to accept the conclusions without a mountain of evidence across several datasets and settings.

The experiments are very well-structured. Sections 4.1 and 4.2 presented a particularly clear narrative that was easy for me to follow and which was well-supported by results.

The one aspect of the datasets that was hard for me to evaluate were whether LowMismatch, MediumMismatch, and HighMismatch were perfectly representative of the phenomena in question. These are assembled from different "base" datasets. It sounds like they differ primarily in the entity composition and how they are sampled. However, I could imagine there being other subtle differences that lead to the observed results, meaning that the experiment is not perfectly controlled where *only* the distribution over entities/relations is what's varying. Some discussion of this would be helpful.

WEAKNESSES

As I said above, I think the results of this paper could potentially be broader. Frequency Shock is a very interesting phenomenon, related to bias amplification (see, for example, https://aclanthology.org/D17-1323/ ). Are there other related problems where this shows up?  It feels like this paper could potentially be positioned around a stronger claim. However, this is not necessary for TMLR, as we are just evaluating what is claimed here.

The solution this paper presents, combining multiple models, was overall a bit unsatisfying. While it's an appealing and low-hanging solution, I felt like I didn't learn very much from seeing it presented. The mixed training setting (combining with masked LM pre-training) was interesting but the results here weren't all that strong. In general, it felt like solutions from other problems were being tried here, but we weren't learning anything new from this problem specifically.

---

> ### Author Response · Authors · 2023-02-21
> **Author Response**
>
> We thank the reviewer for constructive feedback. We hope that our revised version and the following responses address your comments satisfactorily, and we would be happy to discuss and address any other concerns.
>
> > The one aspect of the datasets that was hard for me to evaluate were whether LowMismatch, MediumMismatch, and HighMismatch were perfectly representative of the phenomena in question. These are assembled from different "base" datasets. It sounds like they differ primarily in the entity composition and how they are sampled. However, I could imagine there being other subtle differences that lead to the observed results, meaning that the experiment is not perfectly controlled where only the distribution over entities/relations is what's varying. Some discussion of this would be helpful.
>
> Although our datasets are based on other “base” datasets from the literature, note that these datasets are not completely independently constructed. LPAQA was created to carefully follow the characteristics of LAMA (see Section 5.1 of [1]) so it can be used as a training set (it is now widely adopted by the community as the training set); we also use it in our LowMismatch dataset for the same purpose and in a similar way as existing literature. LANKA was also created to carefully follow the characteristics of LAMA (see Appendix A of [2]) except for some count statistics to reveal the inherent bias of LMs towards some entities; we indeed use it in our MediumMismatch dataset for its frequency statistics differences and otherwise similar characteristics.
>
> [1] How Can We Know What Language Models Know?
>
> [2] Knowledgeable or Educated Guess? Revisiting Language Models as Knowledge Bases
>
> > Frequency Shock is a very interesting phenomenon, related to bias amplification (see, for example, https://aclanthology.org/D17-1323/ ).
>
> Thanks for bringing this work to our attention. It is indeed quite relevant to our work and points out interesting future directions. In the revised version, we added a discussion section where we discuss the connection between our work and the line of work suggested by the reviewer.
>
> > In general, it felt like solutions from other problems were being tried here, but we weren't learning anything new from this problem specifically.
>
> First, we note that finding simple solutions for Frequency Shock that are easy to implement and maintain is particularly critical for this problem. Since our setting is an OOD scenario, typically there would be limited opportunities to robustly evaluate a significantly different model (such as a new architecture) on the target domain/distribution before deployment.
>
> Second, we summarize some key insights about the problem/solution combination revealed by our analysis (beyond the headline results of boosts in performance):
> *  Figure 3 provides the reason for the boost we obtain by the two solutions, by showing that they indeed fix the frequency shock.
> *  In Table 4, we show that gains from our two strategies come about by re-balancing the accuracies between the rare and common entities.
> *  According to Table 4, Model mixing has a one-sided alleviation effect while mixture training has a double-sided effect (see Section 5.1 for details).
> *  An interesting finding is that high percentages of pretraining task queries are needed in mixture training to prevent frequency shock, see discussion in section 5.2.

---

### Author Response · Authors · 2023-03-02
**Summary of the Discussions**

With the 2-weeks discussion period coming to an end soon, we wanted to take this opportunity to thank the reviewers one more time. Thanks to your valuable feedback, we believe our submission has improved in many ways. A high-level summary of the main improvements made since the first submission can be found under “Changes Since Last Submission”.

We tried to address the concerns of all reviewers as much as possible in our revision. However, we notice that some partial disagreements with reviewer 6iZL may have still remained. We respect that and highly appreciate the multiple rounds of constructive discussion we had with the reviewer.

We hope that the long list of evidence we provided regarding the (potentially large) audience of our work and the removal of subsection 5.3 (which was pointed out by the reviewer as the remaining blocker for claim/evidence match) are convincing enough for the reviewer to consider changing their scores on the audience and claim/evidence match questions.

---

### Decision · Action_Editors · 2023-04-02

**Recommendation:** Reject

**Comment:**

The problem studied in this work is of great importance in practice and will be interesting to a large portion of the audience in ML community. After reading the paper and review/response discussions, my major concern is about the novelty, soundness, and support of the main findings.

1. It is well known that highly-similar training-test distributions will lead to good test accuracy (finding 1 in Section 6) and dissimilar training-test distributions will lead to poor test accuracy (finding 2 in Section 6).

2. It is not clear whether finding 3 really holds in general given the above two questions raised by Reviewer 6iZL.

3. Main observations/conclusions heavily rely on some implicit assumptions, which are not well justified and questionable.

    a. About the distributions.
    >Let D1 be the frequency statistics of the pre-trained model, D2 be that of the finetuning data, and D3 be that of the finetuned model. If D3 was somewhere between D1 and D2, then we would agree with the reviewer that this would not be surprising and totally expected. However, the surprising part is in the amplification: entities that are frequent in D1 and less frequent in D2 become extremely infrequent in D3 and entities that are infrequent in D1 and more frequent in D2 become extremely frequent in D3 (see the distributions in Figure 3(a) and Table 5).

    It is surprising to me that D3 is not somewhere between D1 and D2. If it was not, something wrong with the finetuning process? Note that both the two proposed methods, model mixing and mixture finetuning aim to make the final model somewhere between D1 and D2. I suggest the authors provide clear explanations, which will make the claim convincing.

    b. regarding finetuning dataset
    >in our case, however, we aim to retain as much as possible the frequency statistics learned from domain A (i.e. the pre-training task) and avoid adapting to the target domain B (i.e. the finetuning dataset)

    I'm a bit confused: if we aim to retain pre-training information and don't want to adapt to the finetuning dataset, why do we need finetuning? It is better to better support this claim.

While listing the above concerns, this work definitely has the potential to be solid and bring great insights to the community. The authors are encouraged to address those issues and re-submit a new version.

**Audience:**

Yes

**Claims And Evidence:**

The claims are not well supported by accurate, convincing, and clear evidence. I largely agree with Reviewer 6iZL:
> 1. The paper shows frequency shock only on one dataset (MediumMismatch). Is it also true for other datasets?
2. Even the paper does not show clear evidence for MediumMismatch. Only via the following responses (not in the paper), it turns out that the training set contains more common entities than rare ones (435 vs 355). But the drop of common percentage in fig 3a could be due to different reasons rather than frequency shock; for instance, due to the context given to the model (e.g. the query template, asked relations).